# Adverse Events of Radioligand Therapy in Patients with Progressive Neuroendocrine Neoplasms: The Biggest Eastern European Prospective Study

**DOI:** 10.3390/cancers16203509

**Published:** 2024-10-17

**Authors:** Adam Daniel Durma, Marek Saracyn, Maciej Kołodziej, Katarzyna Jóźwik-Plebanek, Dorota Brodowska-Kania, Beata Dmochowska, Adrianna Mróz, Beata Kos-Kudła, Grzegorz Kamiński

**Affiliations:** 1Department of Endocrinology and Radioisotope Therapy, Military Institute of Medicine—National Research Institute, 04-141 Warsaw, Poland; msaracyn@wim.mil.pl (M.S.); mkolodziej@wim.mil.pl (M.K.); gkaminski@wim.mil.pl (G.K.); 2Department of Endocrinology and Neuroendocrine Tumors, Department of Pathophysiology and Endocrinology in Zabrze, Medical University of Silesia, 40-055 Katowice, Poland

**Keywords:** neuroendocrine neoplasms, neuroendocrine tumors, NEN, RLT, PRRT, 177-Lu, 90-Y, complications, adverse events, CTCAE

## Abstract

**Simple Summary:**

Radioligand therapy (RLT) with the use of [^177^Lu]Lu-DOTA-TATE and [^90^Y]Y-DOTA-TATE is a subsequent line of the neuroendocrine neoplasms (NENs) treatment. A total of 127 patients with progressive NENs underwent 4 courses of RLT. Changes in laboratory parameters, adverse events (AEs), progression-free survival (PFS), and overall survival (OS) were analyzed. RLT caused changes in laboratory parameters regarding kidney and bone marrow function, as well as glucose concentration. Nevertheless, RLT appeared to be a safe method of NEN treatment with a low number of AEs and a high percentage of survival rate in a long-term observation.

**Abstract:**

Background: Neuroendocrine neoplasms (NENs) are neoplastic tumors developing in every part of the body, mainly in the gastrointestinal tract and pancreas. Their treatment involves the surgical removal of the tumor and its metastasis, long-acting somatostatin analogs, chemotherapy, targeted therapy, and radioligand therapy (RLT). Materials and Methods: A total of 127 patients with progressive neuroendocrine neoplasms underwent RLT—4 courses, administered every 10 weeks—with the use of 7.4 GBq [^177^Lu]Lu-DOTA-TATE or tandem therapy with 1.85 GBq [^177^Lu]Lu-DOTA-TATE and 1.85 GBq [^90^Y]Y-DOTA-TATE. Assessment of short- and long-term complications, as well as the calculation of progression-free survival (PFS) and overall survival (OS) were performed. Results: RLT caused a statistically but not clinically significant decrease in blood morphology parameters during both short- and long-term observations. Glomerular filtration rate (GFR) significantly decreased only in a long-term observation after RLT; however, it was clinically acceptable. Computed predictions of progression-free survival (PFS) and overall survival (OS) indicated that five years post-RLT, there is a 74% chance of patients surviving, with only a 58.5% likelihood of disease progression. Conclusions: Computed predictions of PFS and OS confirmed treatment efficiency and good patient survival. RLT should be considered a safe and reliable line of treatment for patients with progressive NENs as it causes only a low number of low-grade adverse events.

## 1. Introduction

Neuroendocrine neoplasms (NENs) are uncommon neoplastic entities most often originating within the gastrointestinal tract, particularly in the stomach, small intestine, and pancreas [1,2]. These primary tumor locations classify neoplasms as gastroenteropancreatic (GEP) NENs [3]. While these tumors are typically present in the midgut-derived locations, they can also emerge in diverse anatomical sites such as the lungs, retroperitoneal space, kidneys, gonads, or even bones, classified then as non-GEP NENs [4,5,6,7,8].

The distribution of certain tumor locations displays geographical variability, which may result from genetic differences as well as local lifestyles [9,10]. The United States databases reported an overall incidence of 2.89 per 100,000 persons in 2001–2015 with a yearly increasing incidence [11]. Databases from developed countries like Japan, Switzerland, Italy, or Norway have also been recently bringing data showing an increasing trend in the incidence of NENs [12,13]. Rectal NENs are most common in Asia while in Europe small intestinal NENs are more prevalent. These population differences were confirmed in studies performed on mixed societies like in United States, where the incidence of NENs among African American people is higher than in Caucasians and Asians [11].

The prevalence of the disease is also increasing yearly [14]. This fact may result from better diagnostic methods, new histopathological protocols, and potential environmental factors [15]. In spite of increasing incidence, due to rapid development of new types of therapies and pharmaceutical agents, patients with NENs can be provided with high-level healthcare and treatment, increasing their survival [16,17].

The fundamental therapeutic approach for both GEP and non-GEP NENs involves surgery of primary and metastatic lesions, followed by a prolonged administration of somatostatin analogs in the case of disseminated disease with somatostatin receptors expression [18,19]. Currently, octreotide and lanreotide are two registered somatostatin analogs available for persistent disease management [20,21]. The usage of those agents is dependent on the confirmation of pathological accumulation of radiotracer in somatostatin receptor functional studies—preferably [^68^Ga]Ga-DOTA-TATE PET/CT—but scintigraphy with ^99m^Tc or ^111^In is also acceptable [22,23,24]. Due to a higher PET/CT resolution, sensitivity and accuracy in even small lesions not visible in scintigraphy can be detected with PET/CT-scans. The limitation of octreotide and lanreotide lies in their tolerance, mostly related to gastropancreatic symptoms and adverse events like gallbladder stones [25].

Subsequent therapy lines for disease progression encompass radioligand therapy (RLT), chemotherapy, or targeted therapy. Standard RLT procedures were presented and adapted inter alia in a NETTER-1 study. This involves the administration of [^177^Lu]Lu-DOTA-TATE at activity of 7.4 GBq in four cycles with 8–12-week intervals [26,27]. Simultaneously, standard long-acting somatostatin analog injections are administered during the intervals between radioisotope cycles. An alternative therapeutic approach, though less favored and not registered, involves tandem therapy with a combination of lutetium and yttrium isotopes—[^177^Lu]Lu-DOTA-TATE/[^90^Y]Y-DOTA-TATE [28,29,30]. Additional therapeutic options for progressing NENs include chemotherapy, targeted therapy with tyrosine kinase inhibitor (TKI)—sunitinib—or a selective mammalian target of rapamycin (mTOR) inhibitor—everolimus—and interferon α [31,32,33]. It is imperative to underscore that chemotherapy is advocated as a multi-drug combination regimen, such as capecitabine with temozolomide (CAPTEM) or leucovorin, fluorouracil, and oxaliplatin combined (FOLFOX), and is more reliable in higher-grade neoplasms (with higher Ki-67 proliferation index) and in progressive cases [34,35].

The most common adverse events correlated with second-line therapies include the deterioration of kidney filtration function, impairment of bone marrow function, as well as method-specific outcomes, e.g., skin and mucous membrane disorders during TKI treatment [36,37,38,39]. Radioligand therapy appeared to be the least burdening method of treatment among second-line options, as serious adverse events are uncommon during the treatment. Nevertheless, possible kidney and bone marrow damage have always to be considered, especially with yttrium administration and those parameters controlled during the therapeutic process. As previous studies were not populational ones, the aim of the study was the analysis of potential radioligand treatment outcomes to assess treatment safety, allowing practitioners and patients with NEN to overcome the fear of radioisotope treatment and provide better results.

## 2. Materials and Methods

Patients with progressive neuroendocrine neoplasms (n = 127) were treated with RLT in Endocrinology and Radioisotope Therapy Department of the Military Institute of Medicine—National Research Institute, Warsaw, Poland. The study was approved by the Ethics Committee of the Military Medical Chamber, Protocol Code 154/17. The approval date was 15 December 2017. The study covered 6 years of observation. In this period of time, patients underwent treatment with the use of lutetium, either as monotherapy (7.4 GBq [^177^Lu]Lu-DOTA-TATE) or tandem therapy with lutetium and yttrium (1.85 GBq [^177^Lu]Lu-DOTA-TATE + 1.85 GBq [^90^Y]Y-DOTA-TATE) preceded with [^68^Ga]Ga-DOTA-TATE PET/CT or [^99m^Tc]Tc-HYNIC-TOC scintigraphy (Appendix A). Inclusion and exclusion criteria are presented in Appendix A. All patients in the study group received four courses of RLT, administered every 10 weeks. Between treatment cycles, somatostatin analogs were administered every 4 weeks—lanreotide in 120 mg or octreotide in 60 mg dose. During hospitalization, standardized intravenous nephroprotection, i.e., an amino acid (AA) infusion was administered. In a concentration of 100 g/L and a volume of 500 mL, AA was administered directly before RLT, with 500 mL during RLT (via a separate vascular port) and 500 of AA mL a day after radioisotope administration. After each course of treatment, patients underwent scintigraphy with SPECT/CT scans (Appendix A). Patients were also invited to a long-term treatment evaluation—the status of 107 patients was known, and 44 patients were admitted for personal assessment. The status of 20 patients was unknown—they did not respond to any contact or did not agree to give answers regarding their health status. Thus, they were excluded from a detailed analysis. For the whole study group (n = 127), we performed an analysis of glomerular filtration rate (GFR), creatinine (CREA), erythrocytes (RBC), leukocytes (WBC), blood platelets (PLT), hemoglobin (HGB), fasting glucose (GLU), and chromogranin A (CgA). Parameters were assessed before the first and fourth course of treatment (Appendix A). The study group did not take other medications that could affect the results.

Patients who reported to the Department for a long-term follow-up (n = 44) were selected, and their results were analyzed separately. Tests were performed before the treatment, before the last course of treatment and in a long-term follow-up (Appendix A)

Moreover, adverse events were assessed in a 5-grade-scale according to Common Terminology Criteria for Adverse Events (CTCAE 5.0) [40].

### 2.1. Statistical Analysis

The statistical analysis was performed using the GNU PSPP (version 1.6.2, 2020) and R (version 4.3.1, R Core Team, 2023). To verify whether the results met the rules of normal distribution, the Shapiro–Wilk test was conducted. Results with a normal distribution were presented as means (M) and standard deviations (SD) and in the case of non-parametrical values, as medians (Med.) and interquartile ranges (IQR). Differences between groups were analyzed using appropriate tests, such as the *t*-Student tests and U Mann–Whitney test (with a previous Levene test used to assess the equality of variances in analyzed groups). A significance level of *p* < 0.05 was adopted. For statistical analysis, only full sets of patients’ data were selected. The long-term analysis also covered selected data of patients; thus, laboratory analysis was not interrupted. The lack of data from 20 patients who were unreachable during the control could influence the calculated PFS and OS; however, the factor of “blue-shadowed areas” in Kaplan–Meier curves takes this variability into account.

### 2.2. Laboratory Tests

Blood samples were collected with BD Vacutainer Tests in the Endocrinology and Radioisotope Therapy Department and analyzed in the Medical Diagnostics Department of the Military Institute of Medicine—National Research Institute, Warsaw, Poland. Creatinine and glucose were analyzed using the Roche Diagnostics Assays (Mannheim, Germany) and Hitachi High-Tech Corporation COBAS c503 PRO Automatic Analyzer (Tokyo, Japan). GFR was measured with the use of the CKD-EPI formula (2021 Creatinine). Chromogranin A (CgA) was measured using the LDN Company ELISA test (Nordhorn, Germany). The sensitivity of the method for this parameter was 1.4 µg/L. The morphology was evaluated using the Sysmex Corporation XN 1000 automatic hematology analyzer (Tokyo, Japan). The reference ranges for the laboratory tests discussed in the paper are presented below in Table 1.

## 3. Results

Before Course I of the treatment in the whole study group (n = 127), the mean GFR was 87.32 mL/min/1.73 m^2^, and the creatinine (CREA) concentration was 0.9 mg/dL. Before Course IV, the mean GFR decreased to 83.20 mL/min/1.73 m^2^, and CREA increased to 0.93 mg/dL. This result confirmed the mean decrease in kidney function (GFR = −4.12; ΔCREA = 0.03), but the results were not statistically significant. In blood morphology parameters in that time, a decrease in RBC (Δ = −0.49 mil/µL), WBC (Δ = −1.83 × 10^3^/µL), PLT (Δ = −47.57 × 10^3^/µL), and HGB (Δ = −0.76 g/dL) was observed, and the results were statistically significant for all morphology parameters (*p* < 0.001). The mean fasting glucose (GLU) concentration increased during the treatment of 0.19 mmol/L, but the results were not statistically significant. The median CgA concentration decreased during the treatment (Δ = −41.7 ng/mL)—33.6% of the initial value and was statistically significant (*p* < 0.001) (Table 2).

In the subgroup of 44 patients who reported for a long-term visit between Course I and Course IV, we also observed a statistically significant decrease in RBC (Δ = −0.56 mil/µL), WBC (Δ = −1.85 × 10^3^/dL), PLT (Δ = −37.73 × 10^3^/µL), HGB (Δ = −0.96 g/dL), and CgA (Δ = −70.2 ng/mL, which was 51% of initial value). Changes in GFR, CREA, and GLU were not statistically significant (Table 3).

Comparing Course IV and the long-term observation results, only GFR decreased significantly (Δ = −5.57 mL/min/1.73 m^2^), while Crea and Glu increased significantly (Δ = 0.09 mg/dL and Δ = 0.55 mmol/L, respectively) (Table 4).

Comparing laboratory results before Course I and in the long-term follow-up, we noticed a significant decrease in GFR, RBC, WBC, PLT, and HGB with an increase in CREA and GLU. The Chromogranin A concentration increased, although results were not statistically significant (Table 5).

During the treatment, no grade 3, 4, or 5 side effects were observed in any of the patients in the study group in terms of kidney and bone marrow function (leukopenia, neutropenia, blood unit count, and anemia). Grade 2 adverse events for kidney function were seen in 14.17% of the patients, grade 2 leukopenia in 8.66%, grade 2 thrombocytopenia in 2.36%, and grade 2 anemia in 0.78%. Grade 1 adverse events were seen for kidney function in 38.58% of the patients, leukopenia in 24.41%, platelets count in 18.89%, and anemia in 11.02%. During the treatment, we even observed an improvement in some of the parameters in some patients (Table 6). In 18/67 (26.87%) patients, GFR values increased, while in 10/67 (14.93%), the GFR remained stable. In 3/15 (20%)—hemoglobin concentration and in 15/52 (28.85%)—leucocytes count also increased. In the long-term observation (n = 44), the number of renal AEs according to CTCAE increased to 68.18% (also one to Grade 3), to 43.18% for the platelets count, and to 45.45% for anemia (Table 7).

Directly after the treatment, complete regression (CR) was noted in 1 patient of the whole study group (n = 127) (0.8%), partial regression (PR) in 33 patients (25.98%), stabilization (SD) in 85 patients (66.92%), and progression (PD) in 8 patients (6.3%).

Due to the low death rate of patients in the whole study group (<50%), precise estimation of median OS was unfeasible. Nevertheless, a calculated PFS and OS probabilities, numbers at risk, and Kaplan–Meier estimators are presented in Figure 1 and Figure 2 and Table 8 and Table 9. Data obtained during this estimation proved a high-potential patient survival. Moreover, in the analysis of the representative “follow-up” subgroup (n = 44) directly after RLT, a partial regression was confirmed in 12 (27.27%), disease stabilization in 29 patients (65.91%), and progression only in 3 patients (6.82%)—which clearly corresponded to the results obtained for the whole study group mentioned above.

In the long-term observation (n = 44) compared to the status directly after the treatment, the median observation time was 33.0 months (IQR 33.5). In that period, the neoplastic disease remained stable in 28 patients, and no progression was detected. The observation time of this subgroup was 38.6 months, IQR = 20. Nevertheless, progression was confirmed in 16 patients who previously remained stable (median PFS in this subgroup was 36.0 months; IQR = 30.7).

## 4. Discussion

The study, conducted on one of the largest groups of patients with progressive neuroendocrine tumors treated with the radioligand therapy using lutetium and yttrium isotopes, showed that concerns about potential adverse effects of the therapy may be excessive. Despite achieving statistical significance, the decline in renal parameters did not lead to impairment of kidney function and significant clinical symptoms in the study group. However, radioligand therapy has shown potential for damaging bone marrow hematopoietic function, both in short- and long-term observations. The improvement in peripheral blood parameters in the long-term observation, compared to the results of the last course of treatment, may indicate partial marrow regeneration. We also did not observe severe adverse events in the bone marrow parameters, which aligns with the majority of previously published data.

Chromogranin A levels decreased during the treatment, even in patients with initially normal values for this parameter. The gradual increase in the CgA concentration in the long-term observation may result from the return of secretory function of tumor cells or the manifestation of a renewed growth of the lesions.

An interesting observation was noted in the glucose changes, which did not significantly increase during the treatment, but significantly increased in the long-term observation. Such a result may indicate the potential for long-term changes induced by radioisotope therapy. Due to the increasingly longer survival of patients and growing occurrence of cancer in a younger population, this may suggest the need to observe glucose metabolism disorders to avoid complications related to hyperglycemia.

The relatively low number of observed complications may also result from the proper application of nephroprotection during radioisotope administration, as well as frequent patient monitoring focused on their general internal medicine issues. The study results also show that in some patients, improvement in renal parameters and hematopoietic function may occur. This is most likely due to the aforementioned improvement in their overall condition and a reduction in proinflammatory factors secreted by cancer cells. The positive treatment effect, understood as regression or stabilization of the disease, occurred in over 90% of the patients in the study. The long-term observation—over two years after the end of therapy—proved that stabilization occurred in almost 66% of the patients in the study group. Calculated with the use of statistical software, the percentage of NEN patient survival also showed a good life extension after the RLT.

The registration study of the only currently available original radiopharmaceutical product containing lutetium-177 isotope (Lutathera^®^)—the NETTER-1 study—demonstrated results similar to those obtained in our group. In a group of 111 patients undergoing radioisotope treatment, the median observation time for the group undergoing Peptide Receptor Radionuclide Therapy (PRRT) was 76.3 months in this case. In the NETTER-1 study, the secondary endpoint of overall survival assessment was also not achieved. The estimated median overall survival was 48.0 months in the [^177^Lu]Lu-DOTA-TATE group, compared to 36.3 months in the control group treated with octreotide at a dose of 60mg per month. Additionally, the estimated percentage of patients alive after five years was assessed at 44%. During the long-term observation of patients treated with PRRT, serious adverse events of grade 3 or worse incidents related to the treatment were recorded in 2.7% of the patients in the [^177^Lu]Lu-DOTA-TATE group, and no new serious adverse events related to the treatment were reported after the completion of therapy. In 1.8% of the patients who received [^177^Lu]Lu-DOTA-TATE, myelodysplastic syndrome developed. No new cases of myelodysplastic syndrome or acute myeloid leukemia were reported during the long-term observation. [26,41]

In the study by Kunikowska et al. involving 59 patients treated with 3–5 cycles of combined 1:1 [^177^Lu]Lu-DOTA-TATE and [^90^Y]Y-DOTA-TATE, where the median observation period was 75.8 months, the epidemiological assessment of progression-free survival (PFS) was 32.2 months, and the overall survival (OS) was 82 months [42]. During this period, 42.4% of patients died. In this group, imaging response was confirmed: complete response (CR) in 2%, partial response (PR) in 22%, stable disease (SD) in 65%, and progressive disease (PD) in 6% of patients. The observed 5-year overall survival was 63%, and the 2-year risk of progression was 39.4%. RLT was well tolerated by all patients, with both myelodysplastic syndrome and grade 3 nephrotoxicity affecting 1.7% of patients. No grade 3 or 4 toxicities related to bone marrow or grade 4 impact to kidney function were observed.

In the retrospective analysis by Kennedy et al. involving 104 patients with progressive neuroendocrine neoplasms (NENs) treated with [^177^Lu]Lu-octreotate, the median follow-up was 68 months [43]. The median progression-free survival (PFS) was 37 months, and the median overall survival (OS) was 71 months. The 5-year overall survival was 62%, and the 5-year progression-free survival was 36%. Chronic renal impairment was noted in 1.9% of the patients, and only 1% of the patients developed long-term thrombocytopenia. In this group, there was a relatively high rate of myelodysplasia (MDS)/leukemia (6.7%), possibly due to longer follow-up and the addition of concurrent chemotherapy to 54% of the patients.

In a study involving 62 patients with progressive NENs (GEP–NENs—85.5%; non-GEP-NENs 14.5%) with Grading G1–G3 (G1—33.9%, G2—62.9%, and G3—3.2%) treated with [^177^Lu]Lu-DOTA-TATE, Trautwein et al. showed a partial response (PR) in 16 patients (25.8%), stable disease (SD) in 38 (61.2%), and disease progression (PD) in 7 patients (11.3%) [44]. The 5-year overall survival was 61.8% for all patients, while the 5-year overall survival for non-GEP-NEN was 50.0%, compared to 69.5% for GEP-NEN. Significant predictors of therapeutic outcome were an increase in Chromogranin A (CgA), lactate dehydrogenase (LDH) levels, and patient age. In the study, grade 3 adverse events were observed in 1.6% for platelet count and in 1.6% of patients for leukocyte count.

The utility of CgA as a predictive factor of RLT was also proved in studies by Sharma et al. In a group of 135 patients with a primary tumor location in the small intestine (37.8%), pancreas (26.0%), lungs (13.3%), with unknown primary location (9.6%), or other locations (13.3%), progression-free survival was 23.9 months, and overall survival was 40 months. In 81 patients, pre-treatment chromogranin A was present, and initially higher CgA correlated with lower OS. In patients whose CgA concentration was increasing, the hazard ratio (HR) of death, adjusted for age, also increased [45]. Brabander measured chromogranin A before each therapy and in the follow-up on a group of 354 Dutch patients with GEP-NENs treated with [^177^Lu]Lu-DOTA-TATE. In 74.9% of patients, CgA was initially above the upper normal limit. An increase in CgA by ≥20% compared to the baseline was observed in 76 patients (29%). This increase was present in 34% of the patients who eventually had disease progression and in 27% of the patients who had regression or disease stabilization. In 70% of the patients, CgA returned to baseline levels after therapy [46].

Recent studies have not focused on potential disturbances in glucose metabolism in patients treated with radioligand therapy. During a 2-year-observation after RLT and in a group of 79 patients, Teunissen et al. showed a significant increase in HbA1c concentrations from 5.7% to 6.0% [47]. In their FENET-2016 Trail, Urso et al. suggested a possible role for RLT in inducing modifications of neuroendocrine neoplasm cell metabolism. Especially in a group treated with alternated cycles of [^177^Lu]Lu- and [^90^Y]Y-DOTA-TATE, a greater effect on FDG-positive NEN lesions in comparison to [^177^Lu]Lu-RLT alone was noticed. This may indicate that all cells with somatostatin receptors (SSTR) may get their glucose metabolism changed. Pancreatic cells with a high physiological SSTR presence seem to be particularly vulnerable [48]. While a change in glucose metabolism was noted, the exact and accurate mechanism of the process is unclear. It may result from the direct effect of radiation to the pancreas, impairing beta-cell function; possible kidney radiation, impairing glucose filtration; or the combined effect of radiation to peripheral glucose metabolism and chronic use of somatostatin analogs. Due to a dearth in scientific research regarding the kinds of implications of glucose metabolism changes, we suggest that follow-up studies investigating this potential side effect are necessary. The authors of this manuscript have already started a clinical trial directly focused on glucose metabolism alteration in patients treated with RLT, and the results should be presented in future.

## 5. Conclusions

Our study shows that radioligand therapy induced a statistically significant decrease in blood morphology parameters during both short- and long-term observations. However, this change was, in principle, clinically irrelevant, as low-grade adverse events were predominantly observed. Similarly, only a small percentage of low-grade nephrotoxicity was noted.

Computed progression-free survival (PFS) and overall survival (OS) indicated that five years after the RLT, there was a 74% chance of patients still being alive, with only a 58.5% progression prediction. Good patient survival after the RLT should advocate for this type of therapy in earlier treatment lines, before chemo or targeted therapy.

Thus, RLT should be considered a safe and reliable line of treatment for patients with progressive NEN.

## Figures and Tables

**Figure 1 cancers-16-03509-f001:**
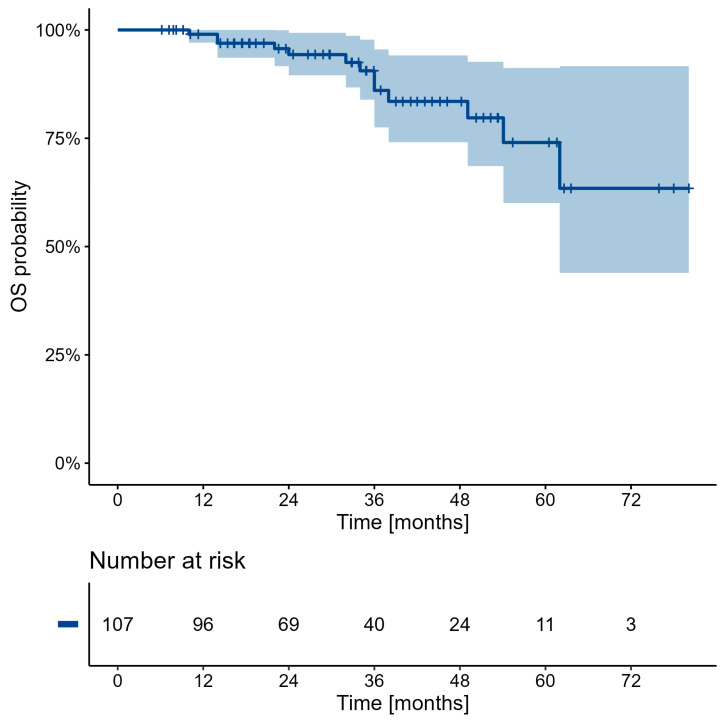
Overall survival probability calculations.

**Figure 2 cancers-16-03509-f002:**
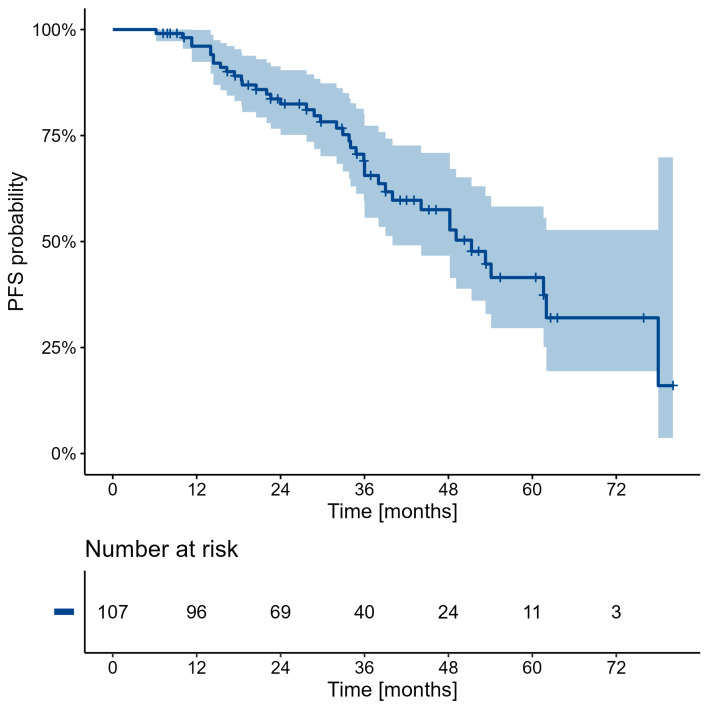
Progression-free survival probability calculations.

**Table 1 cancers-16-03509-t001:** Reference ranges of blood parameters.

Parameter	Unit	Reference Range
GFR	mL/min/1.73 m^2^	>60
CREA	mg/dL	0.7–1.2
RBC	mil/µL	3.5–5.5
WBC	1000/µL	4.0–10.0
PLT	1000/µL	150.0–400.0
HGB	g/dL	11.0–18.0
GLU	mmol/L	3.9–5.6
CgA	ng/mL	19–100

GFR—glomerular filtration rate, CREA—serum creatinine, RBC—erythrocytes, WBC—leukocytes, PLT—blood platelets, HGB—hemoglobin, GLU—glucose, and CgA—chromogranin A.

**Table 2 cancers-16-03509-t002:** Comparison of parameters in the study group (n = 127) before Course I and Course IV of RLT.

n = 127		Before I Course		Before IV Course			CI 95%		
Parameters	Units	Mean	SD	Mean	SD	Δ	L	H	*p*
GFR	mL/min/1.73 m^2^	87.32	22.47	83.20	20.79	−4.12	−11.61	3.37	0.277
CREA	mg/dL	0.90	0.26	0.93	0.26	0.03	0	0.05	0.106
RBC	mil/µL	4.37	0.52	3.87	0.55	−0.49	−0.57	−0.42	<0.001
WBC	1000/µL	6.83	2.29	5.00	2.12	−1.83	−21.4	−1.49	<0.001
PLT	1000/µL	250.25	74.98	202.69	72.44	−47.57	−57.77	−37.37	<0.001
HGB	g/dL	12.92	1.61	12.15	1.55	−0.76	−0.98	−0.55	<0.001
GLU	mmol/L	6.29	2.36	6.48	2.40	0.19	−0.09	0.046	0.185
Parameters	Units	Median	IQR	Median	IQR	Δ	NA	NA	*p*
CgA	ng/mL	124.2	308.5	82.5	244.3	−41.7	NA	NA	<0.001

GFR—glomerular filtration rate, CREA—serum creatinine, RBC—red blood cells, WBC—white blood cells, PLT—blood platelets, HGB—hemoglobin, GLU—glucose, and CgA—chromogranin A. SD—standard deviation, Δ—difference, L—lowest value, H—highest value, CI—confidence interval, *p*—*p*-value, IQR—interquartile range, and NA—not applicable.

**Table 3 cancers-16-03509-t003:** Comparison of parameters in the long-term observation subgroup (n = 44) before Course I and IV of RLT.

n = 44		Course I		Course IV			CI 95%		
Parameters	Units	Mean Δ	SD	Mean Δ	SD	Δ	L	H	*p*
GFR	mL/min/1.73 m^2^	82.14	21.52	82.57	21.5	0.43	−1.88	2.74	0.708
CREA	mg/dL	0.96	0.26	0.96	0.24	0	−0.03	0.03	0.812
RBC	mil/µL	4.36	0.50	3.81	0.54	−0.56	−0.66	−0.45	<0.001
WBC	1000/µL	6.52	1.92	4.67	1.95	−1.85	−2.27	−1.44	<0.001
PLT	1000/µL	222.34	59.79	184.61	67.83	−37.73	−55.82	−19.64	<0.001
HGB	g/dL	12.96	1.25	12.01	1.41	−0.95	−1.25	−0.66	<0.001
GLU	mmol/L	5.98	1.82	6.13	1.39	0.14	−0.52	0.80	0.660
Parameters	Units	Median Δ	IQR	Median Δ	IQR	Δ	x	x	*p*
CgA	ng/mL	137.7	624.5	117.8	563.6	−19.9	NA	NA	0.035

GFR—glomerular filtration rate, CREA—serum creatinine, RBC—red blood cells, WBC—white blood cells, PLT—blood platelets, HGB—hemoglobin, GLU—glucose, CgA—chromogranin A. SD—standard deviation, Δ—difference, L—lowest value, H—highest value, CI—confidence interval, *p*—*p*-value, IQR—interquartile range, and NA—not applicable.

**Table 4 cancers-16-03509-t004:** Comparison of parameters in the long-term observation subgroup (n = 44) before course IV and in the follow-up after RLT.

n = 44		Course IV		Follow Up			CI 95%		
Parameters	Units	Mean	SD	Mean	SD	Δ	L	H	*p*
GFR	mL/min/1.73 m^2^	82.57	21.5	77.00	21.96	−5.57	−9.48	−1.66	0.006
CREA	mg/dL	0.96	0.24	1.05	0.37	0.09	0.03	0.16	0.008
RBC	mil/µL	3.81	0.54	4.05	1.17	0.24	−0.11	0.59	0.175
WBC	1000/µL	4.67	1.95	5.18	2.20	0.51	−0.13	1.15	0.112
PLT	1000/µL	184.61	67.83	183.23	85.45	−1.39	−20.58	17.81	0.885
HGB	g/dL	12.01	1.41	11.82	1.23	−0.19	−0.54	−0.16	0.284
GLU	mmol/L	6.13	1.39	6.68	1.80	0.55	0.14	0.97	0.011
Parameters	Units	Median Δ	IQR	Median Δ	IQR	Δ	x	x	*p*
CgA	ng/mL	117.8	563.6	181.0	1375.7	63.2	NA	NA	0.321

GFR—glomerular filtration rate, CREA—serum creatinine, RBC—red blood cells, WBC—white blood cells, PLT—blood platelets, HGB—hemoglobin, GLU—glucose, and CgA—chromogranin A. SD—standard deviation, Δ—difference, L—lowest value, H—highest value, CI—confidence interval, *p*—*p*-value, IQR—interquartile range, and NA—not applicable.

**Table 5 cancers-16-03509-t005:** Comparison of parameters in the long-term observation subgroup (n = 44) before Course I and in the follow-up after RLT.

n = 44.		Course I		Follow Up			CI 95%		
Parameters	Units	Mean	SD	Mean	SD	Δ	L	H	*p*
GFR	mL/min/1.73 m^2^	82.14	21.52	77.00	21.96	−5.14	−9.49	−0.79	0.022
CREA	mg/dL	0.96	0.26	1.05	0.37	0.09	0.01	0.17	0.038
RBC	mil/µL	4.36	0.50	4.05	1.17	−0.31	−0.69	0.03	0.072
WBC	1000/µL	6.52	1.92	5.18	2.20	−1.34	−2.03	−0.65	<0.001
PLT	1000/µL	222.34	59.79	183.23	85.45	−39.11	−63.90	−14.32	0.003
HGB	g/dL	12.96	1.25	11.82	1.23	−1.14	−1.61	−0.68	<0.001
GLU	mmol/L	5.98	1.82	6.68	1.80	0.70	0.02	1.37	0.044
Parameters	Units	Median Δ	IQR	Median Δ	IQR	Δ	x	x	*p*
CgA	ng/mL	137.7	624.5	181.0	1375.7	43.3	NA	NA	0.658

GFR—glomerular filtration rate, CREA—serum creatinine, RBC—red blood cells, WBC—white blood cells, PLT—blood platelets, HGB—hemoglobin, GLU—glucose, and CgA—chromogranin A. SD—standard deviation, Δ—difference, L—lowest value, H—highest value, CI—confidence interval, *p*—*p*-value, IQR—interquartile range, and NA—not applicable.

**Table 6 cancers-16-03509-t006:** Number of patients who experienced AEs during the treatment due (CECAE 5.0), with additional information regarding change of the initial grade of parameters during the treatment.

n = 127 (100%)	Grade 1	Grade 2	Grade 3	Grade 4	Grade 5	Decrease	Increase	No Change
Kidney Function n = 67 (52.75%)	49 (38.58%)	18 (14.17%)	0	0	0	39	18	10
Leukopenia n = 42 (33.07%)	31 (24.41%)	11 (8.66%)	0	0	0	37	5	0
Platelets Count n = 27 (21.25%)	24 (18.89%)	3 (2.36%)	0	0	0	27	0	0
Anemia n = 15 (11.8%)	14 (11.02%)	1 (0.78%)	0	0	0	12	3	0

**Table 7 cancers-16-03509-t007:** Number of patients with AEs in the long-term observation (CECAE 5.0).

n = 44 (100%)	Grade 1	Grade 2	Grade 3	Grade 4	Grade 5
Kidney Function n = 30 (68.18%)	21 (47.72%)	8 (18.18%)	1 (2.27%)	0	0
Leukopenia n = 14 (31.81%)	9 (20.45%)	5 (11.36%)	0	0	0
Platelets Count n = 19 (43.18%)	18 (40.90%)	1 (2.27%)	0	0	0
Anemia n = 20 (45.45%)	16 (36.362%)	4 (9.09%)	0	0	0

**Table 8 cancers-16-03509-t008:** Overall survival probability (%) of surviving 12, 24, 36, 48, 60, and 72 months with 95% C.I.

Characteristic	12	24	36	48	60	72
Overall	99.0 (97.1, 100.0)	94.3 (89.5, 99.3)	86.0 (77.5, 95.5)	83.5 (74.1, 94.1)	74.0 (60.0, 91.2)	63.4 (43.9, 91.6)

**Table 9 cancers-16-03509-t009:** Progression-free survival probability (%) of surviving 12, 24, 36, 48, 60, and 72 months with 95% C.I.

Characteristic	12	24	36	48	60	72
Overall	96.1 (92.4, 99.9)	82.4 (75.2, 90.4)	65.6 (55.6, 77.3)	57.5 (46.7, 70.9)	41.5 (29.6, 58.3)	32.0 (19.4, 52.7)

## Data Availability

The datasets used and/or analyzed during the current study are available from the corresponding author upon reasonable request.

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
