# Peer review of "Adverse Events of Radioligand Therapy in Patients with Progressive Neuroendocrine Neoplasms: The Biggest Eastern European Prospective Study"

_cancers, 2024, doi:10.3390/cancers16203509_

Round 1

Reviewer 1 Report

Comments and Suggestions for Authors

Title: Adverse Events of Radioligand Therapy in Patients with Progressive Neuroendocrine Neoplasms—the Biggest Eastern-European Prospective Study

Thanks to the authors for this interesting study highlighting the key areas we have to look into during radiotherapy.

My comments:

1. May be I missed it: How was your radiopharmaceutical prepared? What was the % purity and yield. What QC methods were used?

2. Paragraph 86-87: Why was the four courses of the RLT administered at 10 weeks interval? was this based on some previous studies? I'm just curious.

3. Paragraph 86-87: I will advice authors use the term ''administered'' instead of "administrated" (less common term for administer)

4.  Paragraph 214-216: Based on your explanation does it mean RLT might not be effective in the long-term? 

Author Response

Dear Reviewer 1,

First of all, we would kindly like to thank you for the review. We have corrected the manuscript according to all your valuable comments and suggestions. We hope the corrected manuscript will meet all of your expectations. Below, we have attached the answers for all of your questions and suggestions. 

  1. May be I missed it: How was your radiopharmaceutical prepared? What was the % purity and yield. What QC methods were used?

Radiopharmaceuticals of [177]Lu-DOTA-TATE and [90Y]Y-DOTA-TATE were manufactured according to GMP standards for aseptic preparations. Radionuclide precursors of LutaPol and ItraPol (Radioisotope Centre POLATOM, NCBJ) and DOTA-TATE API (Radioisotope Centre POLATOM, NCBJ), both with marketing authorization were used for the production. Each batch of [177Lu]Lu-DOTA-TATE and [90Y]Y-DOTA-TATE was released according to the quality parameter limits of the product determined in the approved quality specifications. The radiochemical purity of [177Lu]Lu-DOTA-TATE and [90Y]Y-DOTA-TATE were tested using HPLC and iTLC with the limitation of  ≥97% in accordance with requirements for therapeutic radiopharmaceuticals. Other impurities, such as DOTA-TATE, metal complexes of DOTA-TATE, and related substances, were determined using HPLC as the sum of the areas of the peaks due to compounds with relative retention with reference to DOTA-TATE of 0.6 to 1.1. Products were tested for pH (limitation: pH 4.0-5.0), bacterial endotoxins (limitation: < 20 EU/mL), and sterility (provided 14 days after product release).

  1. Paragraph 86-87: Why was the four courses of the RLT administered at 10 weeks interval? was this based on some previous studies? I'm just curious.

Long acting somatostatin analogs were being administrated two and six weeks after RLT. Thus 4 weeks after second SSA next RLT was administrated. The study abide the study protocol that was accepted by local ethics committee.

  1. Paragraph 86-87: I will advice authors use the term ''administered'' instead of "administrated" (less common term for administer)

Thank you for the remark. Text was updated.

  1. Paragraph 214-216: Based on your explanation does it mean RLT might not be effective in the long-term? 

The expected median  PFS and OS after RLT in patients with progressive NENs are up to 36 and 48 months respectively. Due to characteristic of the disease, and the fact that other type of treatment (like increased dose of SSA) gives PFS and OS below 12months  we consider RLT very effective. From the NEN patient point of view this additional 2-3 years are crucial and clinically important.

Reviewer 2 Report

Comments and Suggestions for Authors

Neuroendocrine neoplasms are a very interesting topic of medical-oncological and surgical practice, unfortunately they are ubiquitous and we can find them from the skin to the respiratory system to the entire digestive system up to the rectum. A classification defined them Foregut and Mingut. I would remove "neoplastic tumors" from the abstract leaving one or the other. Up until about 10 years ago the fifth and sixth decades of life were the most affected. Unfortunately now they are also diagnosed at younger ages and even, detected in the appendix in pediatric age. Their frequency is increasing. These neoplasms can be detected by chance during diagnostic tests performed for checks for another pathology, Other times they arrive for occlusive or hemorrhagic symptoms and therefore a diagnosis of that disease can be made. Histologically they can be classified as anaplastic NEC, NET of grades 1,2,3; finally there are mixed forms of adenocarcinomas and neuroendocrine neoplasms. Only NETs are able to secrete hormones that can cause symptoms such as skin flushes or repeated episodes of diarrhea. In the majority of cases they do not secrete hormones or they secrete them that are not chemically perfect so they do not have the possibility of binding to the cell membrane receptors. Once the diagnosis has been made With the blood we can measure only chromogranin A, with imaging (double contrast CT, MRI with octreotide in case there are receptors on the cell surface, scintigraphy with Gallium or PET-CT) we can have a lot of information. Where possible the biopsy sample will give us important information such as Ki 67. At this point it will be essential to discuss the case in a multidisciplinary commission to maintain the right therapeutic behavior. Often the surgical act can be important and even definitive. In other circumstances other drugs are at our disposal such as somatostatin and analogues, which still represent the backbone for this pathology. The use of these should be monitored or preceded by cholecystectomy for the complications that they can cause on this organ (PMID: 38051513 to be cited in the bibliography). Everolimus is often proposed (PMID: 29757017 to be cited in the bibliography) with adequate dosage for the side effects on the entire gastrointestinal tract and on the heart. Today there is a tendency to use interferon less. For the most serious neoplasms we agree with what the authors of the paper wrote. Radioligands are drugs that are not easy to handle, generally used in second or third line with positive Gallium scintigraphy, Ki67 less than 55% in the G1 and G2 forms (as, moreover, I believe, is clear from table 6-7 and in the supplementary material) and the patient must be followed by a nuclear medicine doctor. Good work, to be revised, with good iconography, Good bibliography

Author Response

Dear Reviewer 2,

First of all, we would kindly like to thank you for the review, and deep analysis of the presented data. We have updated the manuscript and added visual graphs according to all your valuable comments and suggestions. We hope the corrected manuscript will meet all of your expectations.  

Neuroendocrine neoplasms are a very interesting topic of medical-oncological and surgical practice, unfortunately they are ubiquitous and we can find them from the skin to the respiratory system to the entire digestive system up to the rectum. A classification defined them Foregut and Mingut. I would remove "neoplastic tumors" from the abstract leaving one or the other. Up until about 10 years ago the fifth and sixth decades of life were the most affected. Unfortunately now they are also diagnosed at younger ages and even, detected in the appendix in pediatric age. Their frequency is increasing. These neoplasms can be detected by chance during diagnostic tests performed for checks for another pathology, Other times they arrive for occlusive or hemorrhagic symptoms and therefore a diagnosis of that disease can be made. Histologically they can be classified as anaplastic NEC, NET of grades 1,2,3; finally there are mixed forms of adenocarcinomas and neuroendocrine neoplasms. Only NETs are able to secrete hormones that can cause symptoms such as skin flushes or repeated episodes of diarrhea. In the majority of cases they do not secrete hormones or they secrete them that are not chemically perfect so they do not have the possibility of binding to the cell membrane receptors. Once the diagnosis has been made With the blood we can measure only chromogranin A, with imaging (double contrast CT, MRI with octreotide in case there are receptors on the cell surface, scintigraphy with Gallium or PET-CT) we can have a lot of information. Where possible the biopsy sample will give us important information such as Ki 67. At this point it will be essential to discuss the case in a multidisciplinary commission to maintain the right therapeutic behavior. Often the surgical act can be important and even definitive. In other circumstances other drugs are at our disposal such as somatostatin and analogues, which still represent the backbone for this pathology. The use of these should be monitored or preceded by cholecystectomy for the complications that they can cause on this organ (PMID: 38051513 to be cited in the bibliography). Everolimus is often proposed (PMID: 29757017 to be cited in the bibliography) with adequate dosage for the side effects on the entire gastrointestinal tract and on the heart. Today there is a tendency to use interferon less. For the most serious neoplasms we agree with what the authors of the paper wrote. Radioligands are drugs that are not easy to handle, generally used in second or third line with positive Gallium scintigraphy, Ki67 less than 55% in the G1 and G2 forms (as, moreover, I believe, is clear from table 6-7 and in the supplementary material) and the patient must be followed by a nuclear medicine doctor. Good work, to be revised, with good iconography, Good bibliography

Reviewer 3 Report

Comments and Suggestions for Authors

This study provides valuable insight into the adverse events and efficacy of radioligand therapy for progressive neuroendocrine neoplasms in a large Eastern-European cohort. It is well-organized but could benefit from deeper exploration of certain aspects and the addition of visual aids to enhance the presentation of the results.

Introduction: I suggest to Include more recent literature, especially global data on NEN trends, to provide broader context and enhance the relevance of the study.

Methodology: I suggest to provide more clarity on how confounders were accounted for in the statistical analysis. Discuss the implications of the 20 missing patients in terms of potential bias or effect on results.

Results: I suggest to include additional visual aids, such as survival curves and graphs showing trends in adverse events and blood parameters. Provide more discussion on the significance of glucose changes and their potential long-term clinical implications. I suggest to explain the abbreviations in the tables.

Discussion: I suggest to expand the discussion on the implications of glucose metabolism changes and suggest follow-up studies to investigate this potential side effect. 

Author Response

Dear Reviewer 3,

First of all, we would kindly like to thank for the review. We have corrected the manuscript according to all your valuable comments and suggestions. We hope the corrected manuscript will meet all of your expectations. Below, we have attached the answers for all of your questions and suggestions.

Introduction: I suggest to Include more recent literature, especially global data on NEN trends, to provide broader context and enhance the relevance of the study.

The text and references was updated.

Methodology: I suggest to provide more clarity on how confounders were accounted for in the statistical analysis. Discuss the implications of the 20 missing patients in terms of potential bias or effect on results.

The Statistical analysis in Material and methods was updated.

Results: I suggest to include additional visual aids, such as survival curves and graphs showing trends in adverse events and blood parameters. Provide more discussion on the significance of glucose changes and their potential long-term clinical implications. I suggest to explain the abbreviations in the tables.

Additional graphs as well as abbreviations below tables were added.

Discussion: I suggest to expand the discussion on the implications of glucose metabolism changes and suggest follow-up studies to investigate this potential side effect. 

Discussion was updated.

Round 2

Reviewer 3 Report

Comments and Suggestions for Authors

The authors improved significantly their article.